# Traditional Diet and Environmental Contaminants in Coastal Chukotka IV: Recommended Intake Criteria

**DOI:** 10.3390/ijerph16050696

**Published:** 2019-02-27

**Authors:** Alexey A. Dudarev, Sveta Yamin-Pasternak, Igor Pasternak, Valery S. Chupakhin

**Affiliations:** 1Department of Arctic Environmental Health, Northwest Public Health Research Center, 191036 St-Petersburg, Russia; valeriy.chupakhin@gmail.com; 2Institute of Northern Engineering and Department of Anthropology, University of Alaska, Fairbanks, AK 99775, USA; syamin@alaska.edu; 3Institute of Northern Engineering and Department of Art, University of Alaska, Fairbanks, AK 99775, USA; gipasternak@alaska.edu

**Keywords:** subsistence food, traditional diet, Indigenous people, cuisine, aesthetics, environmental contaminants, PTS, POPs, DDT, PCB, metals, Hg, Pb, As, food safety limits, TDIs, ADIs, Recommended Food Daily Intake Limits, coastal Chukotka, Russian Arctic

## Abstract

The article is the last in the series of four that present the results of a study on environmental contaminants in coastal Chukotka, conducted in the context of a multi-disciplinary investigation of Indigenous foodways in the region. The article presents the Recommended Food Daily Intake Limit (RFDIL) guidelines of the locally harvested foods in coastal Chukotka. The guidelines were developed based on the results of the analysis of the legacy persistent organic pollutants (POPs) and metals found in the samples of locally harvested food, which was collected in 2016 in the villages of Enmelen, Nunligran, and Sireniki on the south coast of the Chukchi Peninsula, Russian Arctic. The overall aim of the article is to expand the toolset for dealing with the challenges of: (1) setting the dietary recommendations when we assess multiple contaminants in a variety of foods (and our method of RFDILs calculation is an example of a possible approach), and (2) managing the real-life circumstances when many types of foods are mixed in many dishes regularly and the concentrations of contaminants in these mixed dishes become uncertain. Drawing on perspectives from the fields of environmental health sciences, humanities, social sciences, and visual art, the authors consider the RFDILs of the examined foods in the context of the culinary practices and aesthetics values (those that relate to the culturally held ideas of beauty ascribed to a dish or the processes of its preparation and consumption) of the Indigenous Arctic cuisine in the region of the Bering Strait, and in the broader dynamics of food and culture.

## 1. Introduction

This article is the fourth in the series of four featured in the issue. The previous three articles discuss the dietary pattern, legacy POPs (persistent organic pollutants), and metals in the locally harvested foods in coastal Chukotka, respectively. The current article focuses on elaboration of the Recommended Food Daily Intake Limit (RFDIL) guidelines and on the problems associated with practical application of the recommendations that researchers in the fields of environmental health science, anthropology, and aesthetics—all are members of our research team—carry out on the basis of field research conducted in Chukotka and the Bering Strait region. Mitigating the hazards of environmental contaminants in the subsistence food sources for the Arctic people is a longstanding, highly sensitive, and complex challenge. Marine mammals, tundra plants, and other local sources of food are central to Indigenous culture and cuisine, as are the relationships, activities, and knowledge connected with their harvest. Their centrality in the Indigenous ways of life puts the issue of environmental contaminants directly into the convergence point between the realms of health, food security, cultural vitality, and the very existence of the Indigenous Arctic residents as uniquely adapted human communities with distinct culture and sense of self. The simultaneous recognition of the nutritional and cultural importance of the Arctic foods, and of the hazards that their consumption poses due to the presence of environmental contaminants, is often referred to as the Arctic dilemma. Addressing this problem effectively and comprehensively requires mitigation at the global international level with cooperation between policy and industry in a wide range of fields [1]. The efforts that pursue dietary and other adjustments at local levels cannot provide a thorough solution. However, communities and health professionals do search for strategies to help reduce the risk of exposure to the environmental contaminants while retaining the nutritional benefits and vast cultural roles of the Indigenous Arctic foods. Of the growing body of work that speaks to these efforts, the account by Watt-Cloutier [1] is particularly edifying in our view. The experience and perspective it offers are those of a Canadian Inuit scholar and a longtime leader of the Inuit Circumpolar Council that represents the Inuit of Chukotka (speakers of Siberian Yupik), Alaska (Siberian Yupik, Central Yup’ik, Inupiaq), Canada, and Greenland. During the decades she has dedicated to advocating for the regulatory policies to help eliminate or reduce the release of environmental pollutants that affect food safety and health in the Arctic, Watt-Cloutier stayed in continuous conversation with Inuit mothers and hunters, discussing the entanglements of food and culture, food security, health, and wellbeing. Overwhelmingly, the perspective of the Inuit community sees the benefits of the Indigenous foodways to outweigh the potential harm. The emphasis throughout is that, while the anti-pollution legislation is to remain a long-term goal, the immediate critical need is to look for strategies to sustain the Indigenous Arctic foodways while minimizing the exposure to environmental contaminants.

The conviction that locally harvested foods are healthier than most potential store-bought alternatives, discussed by Watt-Cloutier [1], is also strongly pronounced in the Indigenous communities of the Bering Strait. In Alaskan and, increasingly in Chukotka, people have negative associations with the prefabricated “processed” foods. On multiple occasions while discussing the subject, one of our hosts would take a packaged store-bought item out of a pantry and point to the colorants and preservatives on the ingredients list, which highlights that none of those are present in the local food. People emphasize that they do not know about the raising, processing, and transporting conditions to which the chicken, pork, and beef, sold (when available) in the village store, have been subjected. They often point to the coloration indicating the product has “freezer burn” and history of several instances of thaw and re-freezing prior to vending. They make similar remarks about the produce, wondering about “chemicals” used in the cultivation, alongside the observations that fruits and vegetables often do not bare appetizing appearance by the time they reach the shelf of a village store. In Chukotka, the vast majority of food offered at the store show old expiration dates, which range from several weeks to several years. There is a proverb among Chukotka residents, saying that, when shopping for prepackaged foods at the store, one should look for “the mostly freshly expired products.” It is a sarcastic play-on-words that is also meant to be genuine and pragmatic advice. Many residents, especially those who have traveled to central regions of Russia and have seen fresh goods sold at stores at a fraction of the cost, feel they are being duped by dishonest entrepreneurs into paying exorbitantly high prices for the goods that have been discarded elsewhere, because the goods being offered to them are unsellable in the places where the residents are thought to have better choices. In this context, for the Indigenous people of Chukotka, eating what they proudly consider the most nutritiously rich foods, harvested and prepared by them or their relatives and community members, is also an assertion of sovereignty and self-reliance. That sentiment is especially amplified when the store is grossly undersupplied, or when people are lacking the means to make sufficient purchases to provide a meal.

A number of researchers seeking to reduce the hazard of ingesting environmental contaminants through dietary practice also recognize the multi-faceted importance of Arctic Indigenous foods and the potential harm that can result from ill-advised avoidances or abandonment of a traditional diet. In their seminal article on mitigating the hazard of methylmercury in the dietary sources of Canada’s First Nations, Wheatley and Paradis [2] warn of a devastating impact that blanket recommendations to avoid eating locally caught fish may have for the health and well-being of the Indigenous communities. In the cases they discuss, the consequences included diabetes, obesity, socio-economic damage, violence, substance abuse, and suicide. They conclude that, with the exceptions of where the levels of contamination are extreme, “the potential negative effects of advising changes in traditional eating patterns outweigh the potential risks from methylmercury for most Canadian Aboriginal peoples” [2]. While the above approach entails calculating the risk in order to determine whether or not a population should be advised against eating locally harvested foods, the approach by Calder et al. [3] offers a more nuanced framework, built on trade-off and substitution scenarios, to be considered in the cases where dietary change, due to high contaminant levels, are advisable. Working in the Labrador Inuit region of Canada, and also focusing on methylmercury levels in traditional food sources, the authors [3] introduce a total of five scenarios where the traditional foods of the Labrador Inuit are replaced by: (1) nutrient-dense store-bought food, (2) processed meat, (3) vegetables, (4) snack foods, (5) Atlantic salmon, with the last scenario being identified as the most viable by the Nunatsiavut Government because Atlantic salmon is a preferred food item that is already part of the local diet. 

A gap that we find to exist in this body of work is a consideration of Indigenous foodways not only as an integral part of the social-ecological adaptation, nutrition, nourishment cultural vitality, and community well-being, but also as a domain of everyday practice, where specialized knowledge and skill are applied to create certain experiences—social, aesthetic, sensory, desired on the daily, meal-to-meal basis. Such cravings may be for certain combination of flavors, textures, temperatures, and aromas of foods, which, in turn, work their way into different types of meals that are eaten in a specific way and are prescribed for certain company, time of day, occasion, or season. This domain of practice is generally known as cuisine.

The current body of work (the present series of four articles, featured in this issue, and especially this article and the first one) aims to contribute an approach, where cuisine is one of the parameters to be considered in developing strategies and recommendations for mitigating the exposure to contaminants through the consumption of locally harvested foods.

This article attempts to contribute to the portfolio of such strategies, using the case study of food and culture in the Bering Strait region of the Arctic. We offer insight from the vantage point of health sciences, anthropology, and visual art. In the first article of the series, we provide an overview of the Bering Strait cuisine, which was studied in the course of multi-year ethnographic fieldwork. Our core argument in the current article is that the understanding of the culinary principles and aesthetic values of the Arctic foodways is integral to the development of criteria for the recommended intake of the foods, affected by POPs and metals. In the context that relates to food, we define aesthetics as culturally held ideas of beauty ascribed to a dish or the processes of its preparation and consumption.

## 2. Materials and Methods

### 2.1. Calculation of the Recommended Food Daily Intake Limits (RFDILs)

To calculate the Recommended Food Daily Intake Limits (RFDILs) for the studied food items, we used the established Russian and international values of Acceptable/Tolerable Daily Intakes (ADIs/TDIs) and the highest concentrations of legacy POPs and metals in the analyzed food samples (each specific contaminant in each specific food item) using the following formula: RFDIL (kg ww of food/person/day) = TDI of contaminant (mg/kg bw/day) × body weight (kg)/concentration of contaminant (mg/kg ww of food). All calculations were made for an adult human weighing 60 kg regardless of gender and age.

The allowable daily intakes (ADIs) for the main groups of POPs (dichlorodiphenyltrichloroethane—DDTs, hexachlorocyclohexane—HCHs, chlorobenzenes—HCB and chlordanes—CHL, but not polychlorinated biphenyls—PCBs) are regulated by the Russian standard [4], while Codex Alimentarius [5] disposes the established ADI/PTDI (provisional TDI) for ΣDDT and ΣCHL (equal to Russian ones), but not for HCHs, HCB and PCBs. TDI for PCBs was used from Health Canada, 2004 [6]. Unlike POPs, the metals TDIs are not regulated in Russia. So the TDIs for metals (except Pb) developed by Health Canada, 2010 [7], the UK Environment Agency [8], and the Joint FAO (Food and Agriculture Organization)/WHO JECFA (Joint Expert Committee on Food Additives) [9,10] were used (Table 1).

Evaluations for children or women of child-bearing age demand separate investigations, which were beyond the scope of the present study. For these categories of local population, we used the approach conventionally implemented in the Russian Arctic environmental hygienic studies, when the calculated RFDILs values were reduced by half to get the double decrease of the potential exposure to contaminants [11,12,13]—see the Section 3.2 of the present article.

We did not consider the distribution of body weights because this approach is hardly appropriate for the community-based study where RFDILs were calculated for multiple contaminants (5 POPs groups and 11 metals) in 12 food groups simultaneously. The percentile methodology has not been used in the present study because this methodology is more suitable for the target sub-population studies, such as the eaters of one particular food item when a single contaminant (or a pair) is considered. The main aim of the present research study was to evaluate the RFDILs concerning the “average adult food consumer” in the studied group (without consideration of age and gender). Using the conditionally adopted average weight and the highest concentrations of contaminants analyzed in food items, we suggest the non-exceedance of contaminants TDIs by the overwhelming majority of the community. Principal assumptions and instruments of the present study are similar to that carried out in Alaska by O’Hara team [14,15,16], but the applications of the results are different.

Inferences and practical suggestions for the application of the calculated RFDILs were provided, including the suggested combinations of certain foods to be avoided due to the safety considerations with respect to the contaminant content. Statistical treatment of the data was carried out using the Microsoft Office 2016 software package (Microsoft Corporation, Albuquerque, NM, United States).

### 2.2. Ethnographic Methods

The applicability of RFDILs developed by the environmental health scientists of our team has been evaluated at the backdrop of the culinary and aesthetic practices of the Bering Strait cuisines, which was studied in the course of long-term ethnographic research. The ethnographic insight on the contemporary cuisine in Chukotka and the Bering Strait region, presented in this article, stems predominantly from the data gathered by ethnographers of our team in the course of the current project, which was launched in 2015. Where relevant, we also draw on the findings from the authors’ earlier research in the region, which spans more than three decades. On the whole, our understanding of the Bering Strait cuisine stems from the interactions with hundreds Yupik, Chukchi, and Inupiaq residents in more than 20 communities during fieldwork on both Russian and Alaskan sides of the Bering Strait. The ethnographic field research methods used include participant observation, semi-directed interviews, and the development of a public exhibition on the Bering Strait food ways, which was put together with our community-based collaborators [17].

## 3. Results

### 3.1. Allowable/Tolerable Daily Intakes (ADIs/TDIs) of POPs and Metals and Recommended Food Daily Intake Limits (RFDILs) for the Studied Food Items

Values of the ADIs (TDIs) presented in Table 1 were used for the calculation of the RFDILs for the studied food items, are presented in Table 2.

### 3.2. Inference on Recommended Food Daily Intake Limits (RFDILs)

Due to lipophilic features, the POPs significantly contaminates only marine mammal blubber, while POPs concentrations in fish, marine, and terrestrial mammal meat and fowl are low. Among POPs, the main contaminant that influences the restriction of local food consumption is HCB (for whale blubber and mantak). Chlordanes are relatively high in whale mantak while PCBs have significant concentrations in seals blubber and whale mantak. HCHs and DDTs are low in all studied foods.Among metals, arsenic highly contaminates marine fish, marine mammal blubber, sea weeds, and mussels. Cadmium is high in seafoods (particularly in mussels). Mercury is high in all types of fish and marine mammal meat (particularly in marine fish and seals). Chromium is high in seafoods, aluminum is high in wild plants and seafoods, and manganese is high in berries and wild plants, particularly Rhodiola leaves. Lead, nickel, and barium are low in all studied foods.Marine mammal meat RFDILs are as follows: whale meat is 450 g/day, walrus meat is 300 g/day, and bearded, ringed, and larga seals meat is 230 g/day.Marine mammal blubber RFDILs are as follows: whale blubber and mantak is 180 g/day. Ringed and larga seals blubber and fermented walrus is 70 g/day. Walrus and bearded seal blubber is 50 g/day.Fish RFDILs are as follows: freshwater and migratory fish is 360 g/day while marine fish is 60 g/day.The RFDIL for locally harvested berries tundra is 300 g/day while, for the tundra greens, it is 170 g/day, with the exception of Rhodiola leaves, for which it is significantly restricted to 50 g/day.The greatest extent of restriction concerns seafood (20 g/day), due to high concentrations of As in seaweed, Cd in mussels, and Al in ascidians.The consumption of meat of terrestrial mammals (reindeer and hare) and fowl (goose) does not need to be restricted for consumption (>500 g/day/person could be eaten), based on the concentrations of all POPs and metals investigated.Due to the lack of samples available at the time of the expedition, the present study was not able to establish RFDILs for the viscera of marine and terrestrial mammals (including liver and kidneys, which might have high levels of some metals) and for the different species of fowl consumed throughout the year.It is important to remember that, for small children and for women who are pregnant, plan to become pregnant, or are breastfeeding, the calculated RFDILs values should be reduced by 50%.

## 4. Discussion

One of the goals of our work and the present article is to contribute to the toolbox that researchers use to mitigate human exposure to contaminants in the Arctic (and possibly other regions) through the dietary practices based, in great part, on the consumption of locally harvested food. In the set of recommendations aiming toward a similar goal, Furgal et al. [18] discuss several areas of disconnect, found to be taking place regularly in communication between researchers and communities. The prominent areas of disconnect they point out are in understanding how environmental contaminants are manifested. Risk and risky behavior are culturally shaped concepts in which the criteria differ vastly between university researchers and the hunters, who target some of the largest animals in maritime Arctic, under the conditions of some of the extreme weather in the world. In the ways that communication itself is conducted, typically being more unilateral and direct on behalf of the researchers, whereas, in Indigenous communities, it unfolds in the form of the network sharing characteristic of the Arctic Indigenous cultures. In this case, we suggest that another area of disconnect may be in the researcher understanding the culinary principles guiding the Indigenous cuisine. Among those are meal structure, consumption patterns, and how the different ingredients are prepared and combined in order to achieve the flavors, aromas, textures, and the multi-sensory appeal that pleases the gustatory, olfactory, ocular senses and cravings (see article I in this issue, which offers an overview of the core practices of cuisine for our study region). We are finding that, to date, even those studies of environmental contaminants in the circumpolar Arctic that do consider certain aspects of food and cooking (such as the comparative analysis of contaminants and nutrients in certain marine mammal products and seafood consumed in Northwest Alaska [14,15,16] and the Northern Fish Nutrition Guide for the James Bay Region [19], developed collaboratively with Inuit communities and Indigenous organizations) tend to focus on the individual sources of the locally harvested food, rather than cuisine. In other words, while offering information on contaminants and nutrients in a particular species of fish and ways in which the given fish can be smoked, dried, or baked, the guide does not delve into the web of social and aesthetic relationships that is cuisine. We, on the other hand, posit that joining the perspectives of anthropology and art in the study of cuisine, as attempted in the current study, holds a promise of insight into the types and times of meals. Seasons and social occasions are associated with specific foods, formal elements of specific dishes (like other forms visual art, edible creations also follow principles of harmony, variety, proportion, and balance in arranging the line, shape, space, texture, color, and value elements of a dish), the etiquette of eating specific dishes, and sequencing of the sensory experiences (such as, for example, starting the meal with dipping small pieces of dry meat or fish in seal oil, combining pieces of hot-temperature boiled meat with pieces of fresh-frozen meat, and taking sips of hot meat broth at the end of the meal). In the current body of work, we build on the literature that focuses on Arctic foodways [20,21,22,23,24,25] and the wider approaches to the study of cuisine that are prominent in food anthropology [26,27] in order to take a step toward the latter. While the insight we present here focuses on coastal Chukotka, part of our intention is to encourage multi-disciplinary collaboration that synergizes the research methods and analytical tools of humanities and sciences to deepen the understanding of the dietary practices and patterns. The aspects of those that are more and less likely to be subjected to a recommended alteration, and possibly of the specific culinary mechanisms through which certain desired effects—reducing the exposure to environmental contaminants in food—can be attempted. Overall, we like to suggest that researchers add a broad-based, holistic understanding of cuisine to the existing portfolio of considerations such as cultural understanding of risk, communication issues, and social implications of dietary change championed by researchers working in other regions of the Arctic. We believe that the earlier discussed strategy developed by Calder et al. [3] to mitigate the contaminant exposure of the Labrador Inuit by substituting the higher methylmercury level of locally harvested foods with Atlantic salmon, also stands to benefit from a more thorough investigation of the culinary principles and other practices of the Labrador Inuit cuisine.

The RFDIL guidelines presented below have been developed by the environmental health scientists of our team, are intended to minimize the health risks associated with the exposure to the legacy POPs and metals, found in the sampled locally harvested foods in our study region. These recommendations are based on the calculations, which consider the contaminant content in the examined food in the context of Russian national and international regulations. In turn, the ethnographers on our team, specializing in anthropology and aesthetics of the Bering Strait foodways contemplate the extent to which the said guidelines can be effectively implemented. Together, we also make suggestions on further applications of our study’s findings, using the multidisciplinary perspectives that emerged in the course of our joint work in the Bering Strait.

### 4.1. Environmental Health Science Perspective on the Practical Application Guidelines for Following the Established RFDILs of the Locally Harvested Food

Meat of terrestrial mammals (reindeer and hare) and fowl (goose) may be eaten without limitation every day throughout the year.Freshwater and migratory fish (including all salmon species) may be eaten every day at the portion size not exceeding 360 g/day/person. The only limiting factor for these species is Hg, which is also high in marine mammal meat. Therefore, one should not eat freshwater or migratory fish and marine mammal meat on the same day. It is fine to eat the marine mammal on the day following the consumption of fish, so these two groups of food could be alternated.Marine fish have double the amount of mercury (compared to freshwater and migratory fish) and is highly contaminated by As. It is recommended that not more than 60 g/day/person are consumed, which is much less than the usual portion of food. Therefore, one should not eat marine fish for at least five days following the consumption of 300 g of marine fish. One should also avoid other foods contaminated by As (marine mammal meat and blubber, seaweeds, ascidians, and mussels) during the day marine fish is consumed.Regarding marine mammal meat, only one species should be chosen for consumption on a given day including either 230 g of bearded, ringed, or larga seal meat, or 300 g of walrus meat, or 400 g of whale meat. Special attention should be paid to parallel food eating, namely to avoid synchronous consumption of seafood with any of the marine mammal species (due to As, Hg, and Cr). Tundra greens (due to the aluminum content) should not be synchronized with whale and bearded seal meat.Blubber in all species of marine mammals is contaminated by POPs, with As showing up at the highest level (also compared to the corresponding meat tissues). One type of blubber should be chosen for consumption on a given day: either 50 g of walrus or bearded seal blubber, or 70 g of ringed or larga seal blubber or fermented walrus blubber, or 180 g of whale blubber or mantak. Similar to the recommended intake for marine fish, it is possible to increase the single day consumption of marine mammal blubber to 300–400 g/day/person and then excluding it from the diet for the next few days. Marine blubber should not be paired with marine fish, seaweeds, or mussels. In the instances that these are paired, the quantity of marine mammal meat or ascidians consumed on the same day should be significantly limited (lower than RFDILs).Tundra berries should not be synchronized with tundra greens (due to Mn).Seaweeds, ascidians, and mussels should never be synchronized because of contamination by As, Cd, Cr and Al (some metal concentrations are extremely high). Very low RFDILs (20 g/day/person) may also be exceeded on a given day, pending the subsequent exclusion of these items from the diet for several days.

### 4.2. Ethnographic Insight on the Challenges and Possibilities of Introducing the RFDIL Guidelines

Although the attitudes toward the specific types of food vary complexly based on the number of factors [21,22,23,24], the overwhelming majority of the people in our study region uphold the belief that eating local foods is vital for their physical survival, spiritual well-being, community vitality, and the possibility to exist as a people. They also believe that the entire way of life enveloping the consumption of local food, including the activities of harvesting, processing, preparing, and eating, is at the core of a healthy lifestyle. Subsistence food is valued not only as a product, but as a vehicle for preserving and transmitting the unique expert knowledge and skill, through which people draw connections to their homeland and ancestors based on which they construct their vision of cultural continuity and assess their hopes for the future. The social values and processes of cooperation and sharing play an important role in mediating wellness at individual and community levels. Without that, the hard work of traveling on the tundra and sea, cooperatively harvesting large animals, tending to great quantities of the fish and plant products that need to be put away over the short Arctic seasons, is often too difficult or impossible to perform. Among the people who are aware—with varying degrees of understanding the specifics—of the presence and dangers of the contaminants in the locally harvested foods, the prevailing belief is that the nutritional and other benefits of eating these foods outweigh the concerns for the potential harm. This is a strongly shared value throughout the Indigenous Arctic world even though it should not be taken to suggest that the people are willing to accept the status quo. To the contrary, battling the industries and policies responsible for the introduction of contaminants into their foodways is part of an ongoing struggle for social justice, led by Arctic communities and their representing organizations [1,25].

Through this lens, we are able to see how the earlier stated recommendations to avoid synchronous consumption of certain foods pose a challenge with respect to a number of the beloved recipes in the Indigenous cookery of Chukotka. For example, the dish called *green kasha* is a mash of several different tundra greens with the main one being sourdock (*Rumex arcticus)*, mixed with leaves of *Rhodeola rosea*, willow (*Salix arctica*)*,* mountain sorrel (*Oxyria digyna*), and others, depending on the availability and preference. It is prepared by hand, where one continuously kneads the mixture on a wooden platter, while adding oil rendered from the blubber of bearded seal, walrus, or whale (also depending on availability and preference). From there on, we find further variations. Some chefs prefer to add aged blood of reindeer or marine mammals, some add salmon raw, and some do a combination of the above. *Green kasha*’s culinary function is that of a condiment, which is consumed with a wide array of thinly sliced animal parts—meat, mantak, guts, which may be raw-frozen, parboiled, or fermented. As in the case with most foods made of animal flesh, the parts are cut into bite-size pieces and are eaten with fingers, which was used to dunk each tidbit (sometimes a combination of meat and mantak) into this mash-like dip (Figure 1 and Figure 2).

The example discussed above clearly speak to the challenges of introducing the RFDIL guidelines outlined in the previous section, particularly the caution that tundra greens should not be synchronized with whale and bearded seal meat. However, we believe the visual formal elements of the meal structure leave the door open to the possibilities of incorporating the pairings and avoidances suggested in the RFDIL guidelines. Thus, amidst the considerations for nutrition, social relationships, and spirituality embedded within the Bering Strait foodways, we also need to look to the overall aesthetic of the Bering Strait cuisines. The elaborate mosaic-like arrangements, organized by the visual art principles of harmony, variety, balance, proportion, movement, and rhythm, take into account not only the types of food being served but also the visual art elements of line, shape, space, texture, color, and value. These, in turn, connect with the gustatory and olfactory desires that give rise to the culinary strategy of offering a wide array of ingredients, to be ingested in various bite-size combinations throughout the meal. Pending the availability and access to many different types of local foods (a vast food security issue that is beyond the scope of our study), the cuisines of the Bering Strait are far more flexible in allowing for different food combinations, substitutions, and creativities than those of many other culinary traditions.

## 5. Conclusions

Based on our knowledge, this is the only comprehensive study aimed at developing the Recommended Food Daily Intake Limits based on the assessment of multiple legacy POPs and metals in the variety of local subsistence foods from the coastal Chukotka since the beginning of 2000s. Considering the possibilities of reducing the risk of exposure to environmental contaminants through dietary practices, we attempt a more comprehensive take on the local foodways in terms of food sources, culinary practices, and cultural values than recent studies in other circumpolar Arctic regions. We also strive to expand the scope of the important culturally constructed and culturally relative ideas arising at the crossings of Indigenous foodways and environmental contaminants. Thus, the cultural understanding of risk in the contexts of indigenous foodways can now be considered alongside the ideas of food-related aesthetic practices discussed here.

The calculation of the RFDILs for the studied food items is based on the established values of Allowable/Tolerable Daily Intake (ADI/TDI) and on the highest concentrations of POPs and metals in the analyzed food samples. The main limiting factors are: chlordanes, HCB, and PCBs in whale blubber and mantak and seal blubber. As in marine mammal meat and blubber, and in seafood, Cd is in seafood, Hg is in fish and marine mammal meat. Cr is in marine mammal meat and seafood. Al is in marine mammal meat, wild plants, and seafood. Mn is in berries and wild plants.

The results of the contaminant analysis presented in the present series of articles help unveil several concerns that warrant further investigation. Among the most pressing, in our view, is the indication that the legacy POPs, long after their ban, continue to find their way mainly into the marine food chains, and some metals are accumulating in both marine and terrestrial food web, which affects the subsistence species important for the livelihoods of the Indigenous residents of the Arctic. The presence of different contaminants in some species and absence in another species make a compelling case for engaging the traditional ecological knowledge holders and biologists. Together, these differently experienced experts should contemplate the feed base and migration routes of each of the affected and non-affected species, and seeing whether and how the information on the select species feed base and migration routes can help identify the active sources of POPs (both legacy and emerging) and metals.

We must further engage the Indigenous food experts in our host communities, as well as the scholars of fermentation, in shedding light on how the different fermentation processes may affect the specific contaminants. As reported, the present study finds that the fermented walrus blubber generally shows to have half the levels of all POPs, while the levels of HCHs, DDTs, and PCBs found in the samples of home-made alcohol point to the possibility of the fermentation process itself being their source. We must also continue on the path toward identifying the preparation methods and combinations of foods that adhere to the culinary and aesthetic principles of the Bering Strait cuisine, while also helping minimize the risk of exposure to contaminants per the RFDIL guidelines suggested by the environmental health scientists. We believe that capturing the synergy between the culinary principles and aesthetic values employed in the Indigenous Arctic cuisines is a viable and promising task for further collaboration between local food experts and visual artists. In turn, this collaborative aspect should build on this core observation made in the years of our ethnographic research in the Bering Strait. The people in our study region are remarkably generous in sharing knowledge, and they are eager to learn about the food-related practices of others in the communities on the Russian and Alaskan coasts. That, in our view, is part of a prodigious cultural capital, which the environmental health scientists, visual artists, anthropologists, and community-based advisors can mobilize in developing effective materials and programs for educating on contaminants, Indigenous cuisines, and environmental health.

## Figures and Tables

**Figure 1 ijerph-16-00696-f001:**
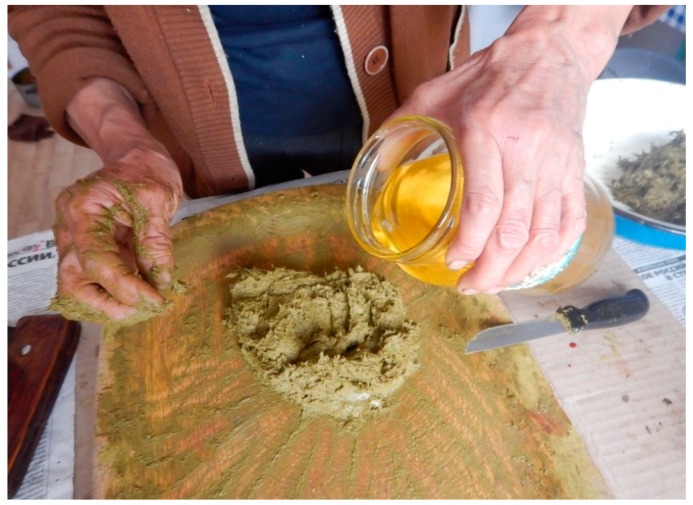
Mixing walrus oil, rendered from blubber, into the plant mix used to make *green kasha*. Photo by Sveta Yamin-Pasternak and Igor Pasternak, 2015. Photo by Igor Pasternak, 2015.

**Figure 2 ijerph-16-00696-f002:**
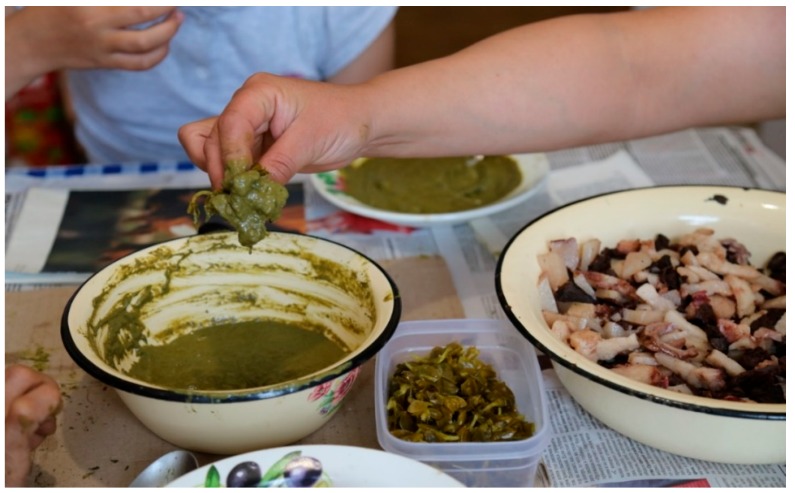
Mealtime: dipping bite-size pieces of parts of walrus and whale, variably combined for each bite, into the *green kasha*. The plastic ware in the middle contains pickled *Honckenya peploides*, which is a locally collected beach green. Photo by Igor Pasternak, 2015.

**Table 1 ijerph-16-00696-t001:** ADIs (TDIs) for some persistent organic pollutants (POPs) and metals used for calculations of RFDILs (Recommended Food Daily Intake Limit).

ADI (TDI)	mg/kg bw /day	Standards, Ref.
ΣDDT	0.01	ADI, Russia, 2013 [4]
ΣHCH	0.01	ADI, Russia, 2013 [4]
HCB	0.0006	ADI, Russia, 2013 [4]
ΣCHL	0.0005	ADI, Russia, 2013 [4]
ΣPCB	0.001	TDI oral, Health Canada, 2004 [6]
Pb	0.0036	TDI oral, Health Canada, 2004 [6]
As	0.003	BMDL0.5 oral, JECFA, 2011 [10]
Cd	0.001	TDI oral, Health Canada, 2010 [7]
Hg	0.0003	TDI oral, Health Canada, 2010 [7]
Cu	0.141	TDI oral, Health Canada, 2010 [7]
Zn	0.57	TDI oral, Health Canada, 2010 [7]
Ni	0.012	TDI oral, UK Env Agency, 2009 [8]
Cr	0.001	TDI oral, Health Canada, 2010 [7]
Al	0.143	TDI oral, calc. from PTWI, JECFA, 2006 [9]
Mn	0.156	TDI oral, Health Canada, 2010 [7]
Ba	0.2	TDI oral, Health Canada, 2010 [7]

ADI (TDI): Acceptable/Tolerable Daily Intakes; DDT: dichlorodiphenyltrichloroethane; HCH: hexachlorocyclohexane; HCB: chlorobenzenes; CHL: chlordanes; PCB: polychlorinated biphenyls; PTWI: Provisional Tolerable Weekly Intake.

**Table 2 ijerph-16-00696-t002:** Calculated Recommended Food Daily Intake Limits (RFDILs) for selected food items for the population of Providensky district of Chukotka.

POPs/Metals	Freshwater Fish	Migratory Fish	Marine Fish	Marine Mammal Meat	Marine Mammal Blubber	Reindeer, Hare, Goose Meat	Mushrooms	Berries	Wild Plants	Sea Weed	Ascidians	Mussels
ΣHCH	NL	NL	NL	NL	NL	NL	nd	nd	nd	nd	nd	nd
ΣCHL	NL	NL	NL	NL	420 g whale mantak	NL	nd	nd	nd	nd	nd	nd
ΣDDT	NL	NL	NL	NL	NL	NL	nd	nd	nd	nd	nd	nd
HCB	NL	NL	NL	NL	180 g whale blubber and mantak	NL	nd	nd	nd	nd	nd	nd
ΣPCB	NL	NL	NL	NL	400 g ringed and spotted seals, 430 g whale mantak	NL	nd	nd	nd	nd	nd	nd
Pb	NL	NL	NL	NL	NL	NL	NL	NL	NL	NL	NL	NL
As	NL	NL	60 g	300 g walrus and bearded seal;	50 g walrus and bearded seal, 70 g ringed and spotted seals and walrus kopalkhen, 300 g whale, 180 g whale mantak	NL	NL	NL	NL	20 g	220 g	90 g
Cd	NL	NL	NL	NL	NL	NL	NL	NL	NL	300 g	300 g	20 g
Hg	360 g	360 g	180 g flounder	450g whale and walrus, 230g bearded, ringed and spotted seals	NL	NL	NL	NL	NL	NL	NL	NL
Cu	NL	NL	NL	NL	NL	NL	NL	NL	NL	NL	NL	NL
Zn	NL	NL	NL	450 g walrus	NL	NL	NL	NL	NL	NL	NL	NL
Ni	NL	NL	NL	NL	NL	NL	NL	NL	NL	NL	NL	NL
Cr	NL	NL	NL	400 g	NL	NL	NL	NL	NL	200 g	60 g	150 g
Al	NL	NL	NL	430 g whale and bearded seal	NL	NL	NL	NL	170 g (120 g Rhodiola leaves)	350 g	20 g	60 g
Mn	NL	NL	NL	NL	NL	NL	NL	300 g	200 g (50 g Rhodiola leaves)	NL	NL	NL
Ba	NL	NL	NL	NL	NL	NL	NL	NL	NL	NL	NL	NL
POPs + metals	360 g	360 g	60 g	230 g bearded, ringed and spotted seals, 300 g walrus, 400 g whale	50 g walrus and bearded seal, 70 g ringed and spotted seals and walrus kopalkhen, 180 g whale, 180 g whale mantak	NL	NL	300 g	170 g; 50 g Rhodiola leaves	20 g	20 g	20 g

NL—no limitation. nd—no data.

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
