# Peer review of "Traditional Diet and Environmental Contaminants in Coastal Chukotka IV: Recommended Intake Criteria"

_ijerph, 2019, doi:10.3390/ijerph16050696_

Round 1

Reviewer 1 Report

The idea of this manuscript, acommodating DRIs and trditional diet, is a good concept; however, this article lacks the description of DRI calculation and the ways to distribute the risk according to the traditional dietary habits. Besides, the paper does not sound like a scentific article. The authors may consider to publish it in another journal.

Author Response

Comments and Suggestions for Authors

The idea of this manuscript, accommodating DRIs and traditional diet, is a good concept; however, this article lacks the description of DRI calculation and the ways to distribute the risk according to the traditional dietary habits. Besides, the paper does not sound like a scientific article. The authors may consider to publish it in another journal.

“this article lacks the description of DRI calculation”

Answer: Lines 49-50. We improved the section Materials and methods and added the formula for calculation of the Recommended Food Daily Intake Limits (RFDIL) which we renamed from DRI for better understanding of the sense of this term.

“…this article lacks the ways to distribute the risk according to the traditional dietary habits.”

Answer: The Discussion and Conclusion sections are almost entirely devoted to the approaches and challenges to addressing the risks of the contaminant exposure according in ways that would align with the traditional dietary habits.

“Besides, the paper does not sound like a scientific article.”

Answer: The reviewer may not be familiar with practices of conducting and reporting multi-disciplinary research, which the current study and manuscript represent, and which by nature are open to a greater variety of scientific narratives than are otherwise found in any one discipline-specific practice of science. The authors have a record of publishing in peer-review science journals with high impact level and have received a number of highly competitive grants, including the one from the US National Science Foundation that supported the current study, in great part precisely due to the merits of the multi-disciplinary collaboration it involves.

Reviewer 2 Report

Introduction

“Multi-vocal” is a strange way of putting it… Do you mean “multi-stakeholder” or something to this effect?

Methods

“All calculations were made for a human weighing 60 kg”

This is probably not the most conservative approach, especially for contaminants with endpoints specific to developing fetuses and children. Usually in deriving recommended intakes, authors consider a distribution of body weights, typical concentrations in food, etc., and then design the recommendations around some percentile (e.g., 90th) for exposure.

Major comment: Are the authors certain that the values they are using in Table 1 are for dietary intakes? For instance, the Health Canada pTDI for Hg is 0.2 µg kg-1day-1for women of childbearing age and children and 0.47 µg kg-1day-1for everyone else based on epidemiology of exposures to methylmercury (the Hg species representing ~100% of Hg in most parts of fish). The authors list 0.3 µg kg-1day-1and cite guidelines for rehabilitation of contaminated sites, which would usually mean possible exposures to inorganic Hg, which has a different toxicokinetic and epidemiological profile. These distinctions are very important.

 I assume the authors mean “hare” (the animal) and not “hair” 

Major comment: I understand that the authors did their own analysis of contaminant concentrations for all these foods, correct? Some description of sampling protocol, analytical methods, etc., is necessary, as is some reflection on how representative that data is of the overall diet of the local population. If the authors did not collect their own data, then some acknowledgment or citation is necessary.

 Discussion

“Freshwater and migratory fish (including all salmon species) may be eaten every day at the portion size not exceeding 360 g/d/person. The only limiting factor for these species is Hg, which is also high in marine mammal meat”

 If Hg is limiting here, why does it not appear in Table 2?

I felt the authors provided a balanced discussion about the applicability of dietary maximum intake values given the competing goal of human nutrition. As the authors point out, traditional foods play in overall dietary sufficiency, especially considering high food prices. I think this article could be strengthened by a more thorough acknowledgment of other work that has been done in this space and an explanation of how these known tradeoffs are handled in the community studied by the authors. For instance, there has been a conversation among indigenous health researchers in Canada for a few decades now about how to balance dietary warnings with the risk that communities will lose faith in their food systems. For example: 

Wheatley, B. and S. Paradis (1996). “Balancing human exposure, risk and reality: questions raised by the Canadian aboriginal methylmercury program.” Neurotoxicology17(1): 241-49.   

Furgal, C., S. Powell and H. Myers (2005). “Digesting the message about contaminants and country foods in the Canadian North: a review and recommendations for future research and action.” Arctic58(2): 103-14.     

In the setting of methylmercury in particular, Hydro-Quebec has done a lot of work developing (and communicating) food consumption advice in close collaboration with indigenous communities so that people view the risks in the context of overall benefits. For instance, the following guide is available in English, French and two dialects of Cree:

Hydro-Québec and Conseil cri de la santé et des services sociaux de la Baie James (2013). “Le guide alimentaire des poissons nordiques. Région de la Baie-James.”

Finally, Calder et al. (2018) have evaluated the actual impact on health endpoints of an indigenous population avoiding increased Hg exposures (and consequently changing their dietary patterns) as compared to accepting increased Hg exposures created by hydroelectric development and found that the overall most health-protective measure would be to double down on the traditional diet, preferentially seeking low-Hg, high-nutrient foods such as salmon. While the scope of that analysis was limited to Hg (as compared to the broad range of contaminants studied by the authors of this paper), the premise of encouraging people not to “avoid” foods but rather to “prefer” the best parts of their existing traditional diet is likely relevant.

Calder, R. S. D., S. Bromage and E. M. Sunderland (2018). “Risk tradeoffs associated with traditional food advisories for Labrador Inuit.” Environ Res 168: 496-506

Given the risk that messaging about the risks of contaminants in local foods may reduce confidence in the safety of traditional diets (regardless of intentions), I wondered how urgent this problem is – for instance, what is the typical intake of these various foods compared to the maxima proposed by the authors? Are they widely exceeded at present day? This would be important context. 

Author Response

Comments and Suggestions for Authors

Introduction

“Multi-vocal” is a strange way of putting it… Do you mean “multi-stakeholder” or something to this effect?

Answer: The commonplace dictionary definition of ‘multi-vocal’ is something that has many meanings and speaks to many kinds of social values. We believe the term is appropriate but it is not essential, we therefore chose to eliminate the use of this word in order to avoid possible confusion among the readers.

Methods

“All calculations were made for a human weighing 60 kg”. This is probably not the most conservative approach, especially for contaminants with endpoints specific to developing fetuses and children.

Answer: this approach is a simple but it is commonly implemented in circumpolar regions and is entirely applicable in the current study. The figure of 60 kg is also recommended in the following manuals in Russia and Belarus:

·         Batyan A, Frumin G, Bazylev V. Fundamentals of General and Environmental Toxicology. Tutorial. Spetslit. 2017. 476 p. (in Russian)

·         Melsitova, I. V. Food quality and safety: a manual. Part 2. Food safety / I. V. Melsitova. - Minsk: BSU, 2016. - 199 p. (in Russian)

Moreover, the figure of 60kg has been in use for many years in the environmental hygienic studies in the Russian Arctic (since the Soviet times) because it realistically accurate. It may be changing during the recent years with the higher consumption of carbohydrates and processed manufactured foods among the Indigenous people of the Arctic; some researchers use the figure of 70kg to represent conventional average weight (e.g. O’Hara et al, 2006; Moses et al, 2009 a,b):

·         Todd O’Hara, Cyd Hanns, Gerald Bratton, Robert Taylor & Victoria Woshner (2006) Essential and non-essential elements in eight tissue types from subsistencehunted bowhead whale: Nutritional and toxicological assessment, International Journal of Circumpolar Health, 65:3, 228-242, DOI: 10.3402/ijch.v65i3.18108

·         Moses, S.K., A.V. Whiting, D.C.G. Muir, X. Wang, T.M. O’Hara. 2009. Organic nutrients and contaminants in subsistence species of Alaska: Concentrations and relationship to food preparation method. International Journal of Circumpolar Health 68(4):354-371.

·         Sara K. Moses, Alex V. Whiting, Gerald R. Bratton, Robert J. Taylor & Todd M. O’Hara (2009) Inorganic nutrients and contaminants in subsistence species of Alaska: linking wildlife and human health, International Journal of Circumpolar Health, 68:1, 53-74, DOI: 10.3402/ ijch.v68i1.18294

The above 3 studies did not use the distribution of body weight or the dietary intake for a child. It was clearly stated (Mosses et al, 2009): “Because the current work was interpreted in terms of a 70-kg male human consumer, the data would need to be re-evaluated for other consumer cohorts, particularly children and women of childbearing age who may have different nutrient requirements or vulnerabilities to contaminants”.

In the present Chukotka study we use a similar approach: our study was interpreted in terms of a 60-kg human adult consumer (regardless of gender and age); evaluations for children and women of childbearing age are beyond the scope of the present study.

Regarding the dietary intake for a child: this topic also demands a separate investigation, but we accentuated in the section 3.2. “Inference on Recommended Food Daily Intake Limits” in the item 10 that “It is important to remember that for small children and for women who are pregnant, plan to become pregnant, or are breastfeeding the calculated RFDILs values should be reduced by 50%”.

Usually in deriving recommended intakes, authors consider a distribution of body weights, typical concentrations in food, etc., and then design the recommendations around some percentile (e.g., 90th) for exposure.

With regard to the distribution of body weights: this approach is hardly appropriate for the community-based studies, particularly for the present study where multiple contaminants (5 POPs groups and 11 metals) in 12 food groups are under assessment for the RFDILs calculation simultaneously. Distribution of body weights is more suitable for the target sub-population studies (e.g. eaters of a single food) when a single (or pair) contaminants are considered.

With regard to the percentiles: we have to apply to the publications cited above (e.g. O’Hara et al, 2006; Moses et al, 2009 a, b) where the authors use the mean concentrations of the contaminants without using the 50th or 95th percentiles.

In our opinion the percentile calculations (e.g., 50th or 95th) are hardly possible and hardly justified for practical purposes, particularly when we are dealing with RFDILs for multiple contaminants (5 POPs groups and 11 metals) in 12 food groups simultaneously. We agree that the recommended approach could be useful in case of a narrow task (e.g. methylmercury in few species of fish), but not for the present study.

In the present study (which is characterized by the breadth of coverage and the unique specificities) the use of the highest contaminant concentrations is an adequate and the only practically implemented way for the calculations of the multiple RFDILs.

Generally speaking, the usefulness of percentile is greatly exaggerated. We have to remember that the percentile is just a number, as well as the mean. The mean displays the center of mass of the sample, while the percentile shows the mark of the upper level of the specified fraction of the sample. The main aim of our research is to evaluate the situation concerning the “average adult representative” in the studied group (without respect to age and gender); here the percentile approach makes no sense, while the conditionally adopted average weight 60kg (mean value) for calculations of the RFDILs based on the highest concentrations of contaminants is really useful. Using the calculated values of RFDILs, obtained by this method, we can be sure of the non-exceedance of TDIs of contaminants by the “average adult representative” of the community. Use of the highest concentrations of contaminants also provides the “covering” of the "high level" consumers, and the compliance of the precautionary principle. In our opinion it is very important result.

 Major comment: Are the authors certain that the values they are using in Table 1 are for dietary intakes? For instance, the Health Canada pTDI for Hg is 0.2 µg kg-1day-1 for women of childbearing age and children and 0.47 µg kg-1day-1 for everyone else based on epidemiology of exposures to methylmercury (the Hg species representing ~100% of Hg in most parts of fish). The authors list 0.3 µg kg-1day-1 and cite guidelines for rehabilitation of contaminated sites, which would usually mean possible exposures to inorganic Hg, which has a different toxicokinetic and epidemiological profile. These distinctions are very important.

Answer: yes, we are sure that the TDI values in Table 1 are ORAL (dietary). In this article we consider inorganic metals, not organometallic compounds, and therefore we use concrete oral TDI for inorganic mercury, which is available in the Guidelines from Health Canada – 2010. Of course we are well aware that methylmercury constitutes majority of total Hg, particularly in seafood. However we did not analyze   methylmercury; we analyzed inorganic mercury along with other 17 inorganic metals (see the Article 3 in the present series of articles). The same question could be raised regarding organometallic compounds of arsenic, nickel, lead, etc. This term is beyond the scope of the present series of articles. And the last argument: 0.3 µg kg-1day-1 (which we use in our calculations) is right between “0.2 µg kg-1day-1 for women of childbearing age and 0.47 µg kg-1day-1 for everyone else”.

I assume the authors mean “hare” (the animal) and not “hair” 

Answer: corrected.

Major comment: I understand that the authors did their own analysis of contaminant concentrations for all these foods, correct? Some description of sampling protocol, analytical methods, etc., is necessary, as is some reflection on how representative that data is of the overall diet of the local population. If the authors did not collect their own data, then some acknowledgment or citation is necessary.

Answer: yes, the reviewer is correct. We are now revising the fourth article in the series “Traditional Diet and Environmental Contaminants in Coastal Chukotka” of four articles. The first one is “Study design and dietary pattern” (already accepted by the journal for the publication), the second one is “Legacy POPs”; the third one is “Metals”. All information on sampling protocol, analytical methods, etc. is provided in the previous articles.

Discussion

“Freshwater and migratory fish (including all salmon species) may be eaten every day at the portion size not exceeding 360 g/d/person. The only limiting factor for these species is Hg, which is also high in marine mammal meat”

 If Hg is limiting here, why does it not appear in Table 2?

Answer: we apologize, we are certain that the Hg line was present when we submitted the manuscript, we believe it was missing in error in the version forwarded to the reviewer; it has been reinserted.

I felt the authors provided a balanced discussion about the applicability of dietary maximum intake values given the competing goal of human nutrition. As the authors point out, traditional foods play in overall dietary sufficiency, especially considering high food prices. I think this article could be strengthened by a more thorough acknowledgment of other work that has been done in this space and an explanation of how these known tradeoffs are handled in the community studied by the authors. For instance, there has been a conversation among indigenous health researchers in Canada for a few decades now about how to balance dietary warnings with the risk that communities will lose faith in their food systems. For example: 

Wheatley, B. and S. Paradis (1996). “Balancing human exposure, risk and reality: questions raised by the Canadian aboriginal methylmercury program.” Neurotoxicology17(1): 241-49.   

Furgal, C., S. Powell and H. Myers (2005). “Digesting the message about contaminants and country foods in the Canadian North: a review and recommendations for future research and action.” Arctic58(2): 103-14.     

In the setting of methylmercury in particular, Hydro-Quebec has done a lot of work developing (and communicating) food consumption advice in close collaboration with indigenous communities so that people view the risks in the context of overall benefits. For instance, the following guide is available in English, French and two dialects of Cree:

Hydro-Québec and Conseil cri de la santé et des services sociaux de la Baie James (2013). “Le guide alimentaire des poissons nordiques. Région de la Baie-James.”

Finally, Calder et al. (2018) have evaluated the actual impact on health endpoints of an indigenous population avoiding increased Hg exposures (and consequently changing their dietary patterns) as compared to accepting increased Hg exposures created by hydroelectric development and found that the overall most health-protective measure would be to double down on the traditional diet, preferentially seeking low-Hg, high-nutrient foods such as salmon. While the scope of that analysis was limited to Hg (as compared to the broad range of contaminants studied by the authors of this paper), the premise of encouraging people not to “avoid” foods but rather to “prefer” the best parts of their existing traditional diet is likely relevant.

Calder, R. S. D., S. Bromage and E. M. Sunderland (2018). “Risk tradeoffs associated with traditional food advisories for Labrador Inuit.” Environ Res 168: 496-506

Answer: The revised version of the manuscript now discusses this literature in considerable detail in the Introduction and Discussion sections; it also explains how the current study uniquely contributes to the body of work that this literature represents.

Given the risk that messaging about the risks of contaminants in local foods may reduce confidence in the safety of traditional diets (regardless of intentions), I wondered how urgent this problem is – for instance, what is the typical intake of these various foods compared to the maxima proposed by the authors? Are they widely exceeded at present day? This would be important context. 

Answer: the structure and average annual consumption of local foods (kg/person/year) is presented in the first article “Study design and dietary pattern” (already accepted by the journal for the publication) of the series; the revised version of the manuscript emphasizes that the current article is the fourth in the series of 4 and, where appropriate, refers the reader to the first article.

Reviewer 3 Report

This paper is a brief report on recommended daily intake of traditional foods in the Bering Straits region. It presents some interesting discussion of the advantages and disadvantages of the RDI model for this region. The paper does not adequately present the methods for assessment of contaminant concentrations in traditional foods, and the dietary analysis is fairly simple. In addition the discussion section is rather limited, and it introduces many concepts that are not fully supported or discussed.

Line 19: clarify what is meant by aesthetic.

Line 27: people of the Arctic , or arctic people. Capitalize Arctic when used as a noun.

Line 28: delete “the before indigenous

Line 30: multifaceted?

Line 33: see line 27: note capitalization rule for the word “arctic” throughout paper.

42: aesthetic needs to be clarified. If the aesthetics are to be discussed, they should be introduced more thoroughly. The cultural importance should also be established, as that is not clear.

49: I believe this means the highest concentration found for a specific compound/element, and not that TDIs were only calculated for the compound/element with the highest concentration on an absolute scale. Clarify.

56: 60kg person. This is a simplistic approach. At minimum, also include dietary intake for a child. Ideally present the distribution.

It is unclear where the underlying data needed to create table 2 came from. It is unclear where the animals are from, how many were sampled,

Also using the highest concentration of the PT while it appears on the face to be health protective, is a simplistic view of this issue. At the very least, RDI for median values should be included as well. Since contaminant distributions tend to be highly skewed, the 95th percentile may also be a better indicator of “high” contamination than the highest sample ever detected.

More clarity on the source of the animal data, and the sample sizes would help with interpretation.

Line 77: was a speciated arsenic analysis done? The organic arsenic species in marine organisms is not as toxic as inorganic arsenic. Again, the source of the underlying contaminant data is important to judge this work.

Line 99: this seems like arbitrary advice. Please provide a citation for this recommendation

Some clarification on the methods of ethnographic research.

Line 218-221: if this is to be discussed, it should be more carefully explained, and the significance of aesthetics to the local indigenous peoples should be demonstrated.

Author Response

Comments and Suggestions for Authors

This paper is a brief report on recommended daily intake of traditional foods in the Bering Straits region. It presents some interesting discussion of the advantages and disadvantages of the RDI model for this region. The paper does not adequately present the methods for assessment of contaminant concentrations in traditional foods, and the dietary analysis is fairly simple. In addition the discussion section is rather limited, and it introduces many concepts that are not fully supported or discussed.

Answer: We are now revising the fourth article in the series “Traditional Diet and Environmental Contaminants in Coastal Chukotka” of four articles. The first one is “Study design and dietary pattern” (already accepted by the journal for the publication), the second one is “Legacy POPs”; the third one is “Metals”. All information on methods, sampling protocol, analytical methods, assessment of contaminant concentrations in traditional foods, dietary analysis, etc. is provided in the previous articles.

Line 19: clarify what is meant by aesthetic.

Answer: The revised manuscript addresses this in lines 22-23 and 50-51.

Line 27: people of the Arctic, or arctic people. Capitalize Arctic when used as a noun.

Answer: We respectfully disagree and adhere to the convention of capitalizing all uses of “Arctic,” noun and adjective that make direct references to the geographic region of the Arctic, which characterizes all instances of use of “Arctic” in this article (as opposed to, for example, a colloquial use of “arctic weather” as a synonym for cold weather that does not directly refer to the geographic region of the Arctic).

Line 28: delete “the before indigenous

Answer: corrected.

Line 30: multifaceted?

Answer: We assume the review suggests substituting multi-vocal with multifaceted. We belive our use of “multi-vocal” is correct. The commonplace dictionary definition of ‘multi-vocal’ is something that has many meanings and speaks to many kinds of social values. We believe the use of the term is fitting because it refers to complex cultural entanglements surrounding food. If necessary for the sake of not confusing unfamiliar readers, in the given sentence the word/adjective “multi-vocal” can be omitted with the rest of the sentence remaining unchanged. We leave it up to the journal editor to make the final decision in this regard.

Line 33: see line 27: note capitalization rule for the word “arctic” throughout paper.

Answer: Please see the earlier response to concern.

Line 42: aesthetic needs to be clarified. If the aesthetics are to be discussed, they should be introduced more thoroughly. The cultural importance should also be established, as that is not clear.

Answer: We believe we have addressed this concern in the revised manuscript by defining this term in the Abstract and Introduction, by explaining that this article is the fourth in the series of 4 presented in this issue and referring the reader to article I for a more thorough introduction of the aesthetic practices surrounding foods in our study region.

Line 49: I believe this means the highest concentration found for a specific compound/element, and not that TDIs were only calculated for the compound/element with the highest concentration on an absolute scale. Clarify.

Answer: of course we used the highest concentration found for a specific compound/element, and then consider the whole list of calculated Recommended Food Daily Intake Limits (RFDILs) for each contaminant (5 POPs groups and 11 metals) in each of 12 food groups.

Line 56: 60kg person. This is a simplistic approach. At minimum, also include dietary intake for a child. Ideally present the distribution.

Answer: this approach is a simple but it is commonly implemented in circumpolar regions and is entirely applicable in the current study. The figure of 60 kg is also recommended in the following manuals in Russia and Belarus:

·         Batyan A, Frumin G, Bazylev V. Fundamentals of General and Environmental Toxicology. Tutorial. Spetslit. 2017. 476 p. (in Russian)

·         Melsitova, I. V. Food quality and safety: a manual. Part 2. Food safety / I. V. Melsitova. - Minsk: BSU, 2016. - 199 p. (in Russian)

Moreover, the figure of 60kg has been in use for many years in the environmental hygienic studies in the Russian Arctic (since the Soviet times) because it realistically accurate. It may be changing during the recent years with the higher consumption of carbohydrates and processed manufactured foods among the Indigenous people of the Arctic; some researchers use the figure of 70kg to represent conventional average weight (e.g. O’Hara et al, 2006; Moses et al, 2009 a,b):

·         Todd O’Hara, Cyd Hanns, Gerald Bratton, Robert Taylor & Victoria Woshner (2006) Essential and non-essential elements in eight tissue types from subsistencehunted bowhead whale: Nutritional and toxicological assessment, International Journal of Circumpolar Health, 65:3, 228-242, DOI: 10.3402/ijch.v65i3.18108

·         Moses, S.K., A.V. Whiting, D.C.G. Muir, X. Wang, T.M. O’Hara. 2009. Organic nutrients and contaminants in subsistence species of Alaska: Concentrations and relationship to food preparation method. International Journal of Circumpolar Health 68(4):354-371.

·         Sara K. Moses, Alex V. Whiting, Gerald R. Bratton, Robert J. Taylor & Todd M. O’Hara (2009) Inorganic nutrients and contaminants in subsistence species of Alaska: linking wildlife and human health, International Journal of Circumpolar Health, 68:1, 53-74, DOI: 10.3402/ ijch.v68i1.18294

The above 3 studies did not use the distribution of body weight or the dietary intake for a child. It was clearly stated (Mosses et al, 2009): “Because the current work was interpreted in terms of a 70-kg male human consumer, the data would need to be re-evaluated for other consumer cohorts, particularly children and women of childbearing age who may have different nutrient requirements or vulnerabilities to contaminants”.

In the present Chukotka study we use a similar approach: our study was interpreted in terms of a 60-kg human adult consumer (regardless of gender and age); evaluations for children and women of childbearing age are beyond the scope of the present study.

Additionally speaking about the distribution of body weights, this approach is hardly appropriate for the community-based studies, particularly for the present study where multiple contaminants (5 POPs groups and 11 metals) in 12 food groups are under assessment for the RFDILs calculation simultaneously. Distribution of body weights is more suitable for the target sub-population studies (e.g. eaters of a single food) when a single (or pair) contaminants are considered.

Regarding the dietary intake for a child, of course, it demands separate investigation, but we accentuated in the section 3.2. “Inference on Recommended Food Daily Intake Limits” in the item 10 that “It is important to remember that for small children and for women who are pregnant, plan to become pregnant, or are breastfeeding the calculated RFDILs values should be reduced by 50%”.

It is unclear where the underlying data needed to create table 2 came from. It is unclear where the animals are from, how many were sampled,

Answer: We are currently revising the fourth article in the series “Traditional Diet and Environmental Contaminants in Coastal Chukotka” of four articles. The first one is “Study design and dietary pattern” (already accepted by the journal for the publication), the second one is “Legacy POPs”; the third one is “Metals”. All information on methods, sampling protocol, analytical methods, assessment of contaminant concentrations in traditional foods, dietary analysis, etc. is provided in the previous articles.

We have reorganized the section Materials and methods where we added the explanatory material and provided the formula for calculation of the Recommended Food Daily Intake Limits (RFDILs) which we renamed (from DRIs) for better understanding of the sense of this term.

Also using the highest concentration of the PT while it appears on the face to be health protective, is a simplistic view of this issue. At the very least, RDI for median values should be included as well. Since contaminant distributions tend to be highly skewed, the 95th percentile may also be a better indicator of “high” contamination than the highest sample ever detected.

Answer: yes, it is a simplistic but absolutely applicable and common approach in the community-based dietary studies, including the circumpolar regions. In some studies (e.g. O’Hara et al, 2006; Moses et al, 2009 a, b) the authors use the mean concentrations of the contaminants without using the 50th or 95th percentiles.

In our opinion the percentile calculations (e.g., 50th or 95th) are hardly possible and hardly justified for practical purposes, particularly when we are dealing with RFDILs for multiple contaminants (5 POPs groups and 11 metals) in 12 food groups simultaneously. We agree that the recommended approach could be useful in case of a narrow task (e.g. methylmercury in few species of fish), but not for the present study.

In the present study (which is characterized by the breadth of coverage and the unique specificities) the use of the highest contaminant concentrations is an adequate and the only practically implemented way for the calculations of the multiple RFDILs.

Generally speaking, the usefulness of percentile is greatly exaggerated. We have to remember that the percentile is just a number, as well as the mean. The mean displays the center of mass of the sample, while the percentile shows the mark of the upper level of the specified fraction of the sample. The main aim of our research is to evaluate the situation concerning the “average adult representative” in the studied group (without respect to age and gender); here the percentile approach makes no sense, while the conditionally adopted average weight 60kg (mean value) for calculations of the RFDILs based on the highest concentrations of contaminants is really useful. Using the calculated values of RFDILs, obtained by this method, we can be sure of the non-exceedance of TDIs of contaminants by the “average adult representative” of the community. Use of the highest concentrations of contaminants also provides the “covering” of the "high level" consumers, and the compliance of the precautionary principle. In our opinion it is very important result.

More clarity on the source of the animal data, and the sample sizes would help with interpretation.

 Answer: Please see the earlier response to concern.

Line 77: was a speciated arsenic analysis done? The organic arsenic species in marine organisms is not as toxic as inorganic arsenic. Again, the source of the underlying contaminant data is important to judge this work.

Answer: In this article we consider inorganic metals, not organometallic compounds, and therefore we use concrete oral TDIs for inorganic mercury, inorganic arsenic, etc. Of course we are well aware that methylmercury constitutes majority of total Hg, particularly in seafood, and that the organic As compounds have different toxicity compared with the inorganic As. However we did analyze inorganic Hg and inorganic As along with other 16 inorganic metals (see the Article 3 in the present series of articles). The term of organometallic compounds is beyond the scope of the present series of articles.

Line 99: this seems like arbitrary advice. Please provide a citation for this recommendation

Answer: in the Russian Arctic environmental hygienic practice this approach is common – we have to reduce the RFDILs by 50% in order to get the double decrease of the potential exposure of children and young women to contaminants. Below are the references:

1.       “Measures for reduction of health risk associated with the contamination by PTS (persistent toxic substances) of traditional food”. Section 8.2 in the Resume of the Final Report “Persistent Toxic Substances, food security and indigenous people of the Russian North”. Arctic Monitoring and Assessment Programme, 2004. Oslo, Norway. 80 pp. ISBN 82-7971-039-6. (in Russian).

2.       Chapter 8. “Measures for reduction of health risk associated with the contamination by PTS (persistent toxic substances) of traditional food”. Additional training program “Basics of life safety in the regions of the Far North”. Methodological manual for schools teachers. Arctic Council Indigenous People’s Secretariat. Copenhagen, Denmark. 2006. 47 pp. (in Russian).

3.       Dudarev, A. A. Dietary exposure to persistent organic pollutants and metals among Inuit and Chukchi in Russian Arctic Chukotka. Int. J. Circumpolar Health. 2012, 71:1, 18592, DOI: 10.3402/ijch.v71i0.18592.

Some clarification on the methods of ethnographic research.

Answer:  A description of the ethnographic methods used is now provided in the Methods section.

Line 218-221: if this is to be discussed, it should be more carefully explained, and the significance of aesthetics to the local indigenous peoples should be demonstrated.

 Answer: This concern has been addressed in the revised manuscript by elaborating on the discussion of cuisine in our study region with additional references to the study of Arctic foodways and the on the more prominent approaches used to study cuisine in the field of anthropology:

Mintz, S. W. and Du Bois, C. The anthropology of food and eating. Annual Review of Anthropology, 2002, 31:1, 99-119.

Sutton,. Food and senses. Annual Review of Anthropology, 2010, 39:209-23.

Kishingami, N. Sharing and Distribution of Whale Meat and Other Edible Whale Parts by the Inupiat Whalers in Barrow, Alaska, USA. 2012. Osaka: Kishigami's Office, National Museum of Ethnology.

Starks, Z. S. Arctic Foodways and Contemporary Cuisine. Gastronomica, 2007 7(1):41-49.

Jolles, C. Z. Faith, Food, and Family in a Yupik Whaling Community. 2002. Seattle: University of  Washington Press.

The current manuscript also explains that this is the fourth in the series of articles I-IV on traditional diet environmental contaminants in coastal Chukotka (at the start of Abstract and Introduction) and by referring to article I in this series that discusses in substantial detail the aesthetic practices connected with foods in our study region (within the vicinity of the lines mentioned by the reviewer). Definition of ‘aesthetic’ in the context of food and cooking is also featured early in the introduction, and in the abstract.

Round 2

Reviewer 3 Report

The text of the manuscript is much improved; however, the dietary intake recommendations are in many cases based on single samples. This is likely to be very inaccurate, due to the high variability of contaminant concentrations among individuals of the same species. There is no effort to discuss this serious limitation.

It is acceptable to present the concentrations of POPS and metals in the previous manuscripts despite low sample sizes, because readers can reasonably be expected to understand the potential for error, and consider the information in reference to other studies. However, basing dietary advice on single samples results in a non-quantifiable error than makes interpretation impossible.

Without recalculation with a larger more reliable dataset (perhaps from a review of other studies) the dietary recommendations in this manuscript should not be published.

Author Response

Dudarev et al,  Response to Review, January 31, 2019

Reviewer comment:

The text of the manuscript is much improved; however, the dietary intake recommendations are in many cases based on single samples. This is likely to be very inaccurate, due to the high variability of contaminant concentrations among individuals of the same species. There is no effort to discuss this serious limitation.

It is acceptable to present the concentrations of POPS and metals in the previous manuscripts despite low sample sizes, because readers can reasonably be expected to understand the potential for error, and consider the information in reference to other studies. However, basing dietary advice on single samples results in a non-quantifiable error than makes interpretation impossible.

Without recalculation with a larger more reliable dataset (perhaps from a review of other studies) the dietary recommendations in this manuscript should not be published

«The text of the manuscript is much improved; however, the dietary intake recommendations are in many cases based on single samples. This is likely to be very inaccurate, due to the high variability of contaminant concentrations among individuals of the same species. There is no effort to discuss this serious limitation.

It is acceptable to present the concentrations of POPS and metals in the previous manuscripts despite low sample sizes, because readers can reasonably be expected to understand the potential for error, and consider the information in reference to other studies. However, basing dietary advice on single samples results in a non-quantifiable error than makes interpretation impossible.

Without recalculation with a larger more reliable dataset (perhaps from a review of other studies) the dietary recommendations in this manuscript should not be published».

Author response:

We thank the reviewer for finding the manuscript to be much improved. Regarding the concern on the sample sizes, we emphasize that our dietary recommendations are based not on single samples results, but on the several samples of each species and each food group; many of samples were POOLED, as indicated in the list below.

·          Freshwater fish: 3 single samples + 1 pooled sample of Arctic char;

·          Migratory fish: 3 single samples of salmon species + 1 pooled sample of humpback salmon;

·          Marine fish: 4 single samples + 3 pooled samples;

·          Marine mammal meat: 3 samples of gray whale + 4 samples of walrus + 4 samples of bearded seal; only ringed seal and spotted seals meat samples were single samples;

·          Marine mammal blubber: 5 samples of gray whale + 6 samples of walrus + 2 samples of bearded seal; only ringed seal and spotted seals blubber samples were single samples;

·          Land mammals meat: 3 samples of reindeer + 3 samples of hare;

·          Mushrooms: 4 pooled samples of 2 species;

·          Wild berries: 7 pooled samples of 3 species;

·          Wild plants: 2 pooled samples of Rhodiola leaves + 1 pooled sample of wild leek + 2 pooled samples of mixed plants;

·          Arctic kelp and blue mussels: both pooled samples;

·          Sea squirts: 4 samples.

We therefore suggest that while our sample size is not large, it is quite representative for the purpose of making dietary recommendations.  

We also must emphasize that the overall aim here is to expand the toolset for dealing with the challenges of 1) setting the dietary recommendations when we assess multiple contaminants in a variety of foods (and here our method of RFDILs calculation is an example of a possible approach), and 2) managing the real-life circumstances when many types of foods are mixed in many dishes regularly and the concentrations of contaminants in these mixed dishes become uncertain.

Our study attempts a comprehensive assessment of potential exposure to contaminants from the locally harvested foods, with the understanding of the role each of the foods, their various combinations, and the patterns and practices connected with their consumption, play in culture and cuisine of the Chukotka Indigenous communities.